# Determining the factors of m-wallets adoption. A twofold SEM-ANN approach

**Imdadullah Hidayat-ur-Rehman** [1]*, **Saeed Alzahrani**[1], **Mohd Ziaur Rehman**[2],
**Fahim Akhter**[1]

1 Department of MIS, College of Business Administration, King Saud University, Riyadh, Saudi Arabia,
2 Department of Finance, College of Business Administration, King Saud University, Riyadh, Saudi Arabia

☯ These authors contributed equally to this work.
* ihidayaturrehman@ksu.edu.sa

**Data Availability Statement:** All relevant data are within the paper and its Supporting Information files.

**Funding:** The author(s) received no specific funding for this work.

## Abstract

M-wallets are comparatively more advantageous and convenient than conventional payment systems as m-wallets allow users to avoid cash. The present research uses the diffusion of innovation theory as the base theory to propose a research model by incorporating constructs like convenience, perceived security, personal innovativeness, and perceived trust to investigate the determinants of consumers' intention-to-use m-wallets. A twofold approach comprising of Structural Equation Modelling—Artificial Neural Network (SEM-ANN) was used: First, partial least squares structural equation modelling (PLS-SEM) was employed to determine the significant determinants of intention-to-use. Second, the ANN approach was applied as robustness to corroborate the outcomes of PLS-SEM and to estimate the relative importance of the SEM-based significant determinants. Our findings confirmed that compatibility, ease of use, observability, convenience, relative advantage, personal innovativeness, perceived trust, and perceived security are the key elements that influence the intention-to-use m-wallets. Moreover, we ascertained that perceived security is the most influential predictor of intention-to-use. The outcomes of ANN have complemented the findings of PLS-SEM, but some differences were also exhibited in the order of influential factors. The study brings to fore significant insights and a set of suggestions for the companies carrying out the development, execution, and marketing of M-wallet services.

## 1. Introduction

Globally, the growing demand for digital transactions has drastically changed the consumers' attitudes about the adoption of mobile payments (MP) systems [1, 2]. The integration of payment methods to mobile technologies has converted the conventional wallets into digital wallets called mobile wallets (m-wallets) [3]. In comparison with other modes of MP, m-wallet is a rather more versatile and new method of online payments [4]. M-wallet replaces a person's conventional wallet by allowing them to save their credit/debit card details to perform financial transactions and store their personal information like office access ID, travel tickets and

**Competing interests:** The authors have declared that no competing interests exist.

insurance policies [5]. M-wallets can be divided into two types i.e. general MP systems and proximity mobile payment systems [6]. Near Field Communication (NFC) is the universally accepted technological standard for general mobile payment methods while the proximity mobile payment systems use Quick Response (QR) code or barcode scanning [7]. Apple Pay, Samsung Pay, and Google Pay are well-known NFC-based mobile payment methods [8].

Teena Wadhera et al. [9] has divided m-wallets into four different categories i.e. semi-closed m-wallets, semi-opened m-wallets, open m-wallets, and closed m-wallets. Semi-closed m-wallets (e.g. Paytm) do not allow any cash withdrawal or redemption and allow only the purchase of products from merchants who have a partnership with the service provider. Semi-opened m-wallets (e.g. Airtel Money) are linked to banks but do not allow cash withdrawal. Open m-wallets (e.g. Vodafone m-pesa) are linked with the banks and allow cash withdrawal at agent outlets or retailers. The closed m-wallets (e.g. Gift Vouchers) allow no cash withdrawal and are non-reloadable with cash. Google Pay, Samsung Pay, Apple Pay, WeChat Pay, Android Pay, Pay Pal, Ali Pay, Paytm, Mada Pay, Oxygen Wallet, STC Pay, Citrus Wallet are examples of m-wallets.

M-wallets have created a competitive business environment for technological companies, financial institutions, and other merchants as m-wallets are attaining considerable market growth due to merchants' realisation about their potential [10]. M-wallets are comparatively more advantageous and convenient than conventional payment systems as m-wallets allow users to avoid cash, facilitate person-to-person remittance transfers and allow remote and proximity payments [11–14].

Although, more than a decade has been passed to the availability of m-wallets in the market, widespread adoption of m-wallets has not been experienced [4]. Consumers are not readily accepting m-wallets despite their benefits and convenience provided to the users [15]. Researchers have mentioned various elements affect the acceptance of m-wallets. Lack of information about the effectiveness of the product, low awareness, privacy norms, innovativeness, resistance, interoperability and infrastructural support are important issues that affect the intention of consumers to use m-wallets [14]. Sharma et al. (2018) applied interpretive structural modelling (ISM) to develop a comprehensive model for mobile wallet inhibitors. According to their results anxiety towards new technology, lack of new technology skills, the complexity of the new technology, and lack of awareness of mobile wallet benefits are the key inhibitors of m-wallets adoption. The low acceptability of m-wallets may be due to the lack of security, trust, awareness, and availability of features [15].

The low adoption of m-wallets has led researchers to carry out various empirical studies. In this regard, most of the studies have been carried out in developed countries. Gao & Waechter [16] studied the impacts of trust on the acceptance of MP systems in Australia. Khalilzadeh et al. [17] studied the NFC-based MP in the context of the United States. Wirth & Maier [18] examined the switching of individuals to mobile payments in Germany. Johnson et al. (2018) explored the limitations to espousal of MP services in the United States environment. Research studies focusing on the acceptance of m-wallets in the Gulf countries is limited [12, 19]. The understanding of m-wallets espousal is important due to many reasons: 1) There are 31.88 million smartphone users in Saudi Arabia in the year 2021 [20]; 2) The number of internet users in Saud Arabia in the year 2021 is 31.91 million [21]; 3) 4G and 5G internet infrastructure is available and about 91% of the population is covered by at least 4G networks [22]. There is an exponential increase in the usage of MP and m-wallets in Saudi Arabia. The total transaction value in the Digital Payments segment of Saudi Arabia is estimated to be US$20,873 million in the year 2021 and mobile point of sale users are expected to reach 8.5 million in 2021 [23]. The payments through the m-wallet segment in Saudi Arabia is expected to increase at a compound annual growth rate of 17.4% during 2018–2025 [24]. The Saudi central bank raised the

monthly e-wallet top-up to 20000 riyals per month in March 2020 to boost digital payment and as an additional precautionary measure against the transmission of the corona virus.

The m-wallet is a new area of research in comparison of other similar domains like mobile banking, e-commerce where extensive research exists in these areas [12, 14]. Moreover, due to the COVID-19 pandemic, digital payment methods are encouraged everywhere. Thus m-wallets provide the retailers more business opportunities if they accept m-wallet payments which will enhance their competitive advantage [25]. However, the adoption of m-wallets is not growing significantly as required [1, 12]. The unavailability of studies about the elements which are critical for consumers' acceptance of m-wallets in the context of developing countries creates a knowledge gap that needs to be focused. Covering this gap is highly important as the adoption of digital payments like m-wallets has major social and economic implications.

In emerging Fintech ecosystems around the world, payments innovation is one of the first frontiers of Fintech innovation. With the government's active support, Saudi Arabia is striving to build a Fintech ecosystem. Fintech innovation is a part of Saudi Vision 2030 to convert the Saudi economic system far from its dependence on oil to a greater technology-driven present-day economic system. The Kingdom has the potential for swift development due to its young, tech-savvy populace and because of government drive of attaiting 70% digital payments transactions by 2030 [26].

Though there are many studies exploring factors that affect M-Wallets adoption in the context of developed countries, there is a lack of study to investigate the factors using M-Wallets in Saudi Arabia. In this regard, we believe that to better understand its potential in Saudi Arabia, it is pertinent to explore the determinants of M-Wallets Adoption in Saudi Arabia through A SEM-Neural Network Approach.

The cardinal novelty of this study contributes to the extant literature is on account of the following reasons:

i. The study employs an SEM-ANN (Structural Equation Modelling–Artificial Neural Network) approach which covers linear and non-linear relationships.

ii. The application of SEM-ANN is applied in an emerging market setting, Saudi Arabia, where no study has been conducted on the factors determining the adoption of M -Wallets

iii. The data was collected during the black swan event, namely, the COVID-19 pandemic situation in Saudi Arabia where the populace was in self-quarantine mode, thus the exploration may be a topical investigation for the greater readership.

The main research question for this study is "What are the key factors which influence the consumers' intentions to use m-wallets?" This study endeavours to achieve the following research objectives:

i. To propose a model comprising of the critical factors that impact the consumers' intentions to use m-wallets.

ii. To make an empirical assessment of the proposed model employing structural equation modelling.

iii. Validating the SEM-based results by using Artificial Neural Network (ANN) analysis.

To achieve the above-mentioned research objectives, this study builds on the prominent diffusion of innovation theory (DOI) [27] by incorporating some important constructs like personal innovativeness, convenience, perceived trust, and perceived security to DOI constructs to propose the model. This study uses the ease of use construct from the technology acceptance model [28] instead of the complexity variable. The DOI is a suitable theory for

describing the innovations' diffusion in social settings contexts [29]. Researchers in the field of information systems have affirmed that the DOI is an appropriate framework that helps in recognising the diffusion of innovations across users having any situation and setting [11]. Previous research using DOI in m-wallets and MP systems settings have confirmed that DOI is the best theory to study m-wallets acceptance [4, 11]. For empirical validation of the proposed model, the partial least squares-structural equation modelling (PLS-SEM) technique was employed. The ANN procedure was employed to confirm the outcomes of PLS-SEM. To attain this purpose, survey data was gathered from m-wallets users in Saudi Arabia, and 737 valid responses were utilised for data analysis. This study offers insights that can help the stakeholders in recognising the influential factors of consumers' adoption of m-wallets. These findings of the study can be leveraged to enhance the attributes of m-wallets and develop strategies to motivate the consumers about m-wallets usage.

The remaining sections of the paper are ordered as follows. Section 2 presents the background and development of the framework. Research methodology is encompassed in section 3. Statistical analysis and results are covered in section 4. The last section highlights the discussion and conclusions. Moreover, this section also incorporates theoretical and practical implications.

## 2. Background and development of the framework

### 2.1 Diffusion of Innovation (DOI) theory

This study used DOI [27] as the base model to explore the elements affecting the intentions of consumers to use m-wallets. Researchers have used DOI as a framework in different disciplines like communications, political science, economics, and information systems and it is considered as a milestone theory because it describes the diffusion process of an innovative product [4, 11]. The diffusion of innovation is propagated in society over time [30]. DOI postulated that innovations having five characteristics i.e. relatively advantageous, compatible, observable, simple, and triable systems are likely to be earlier adopted [30]. Prior research has used DOI to study consumers' behaviours in different contexts like mobile banking, online shopping, autonomous vehicles acceptance and mobile payments [31–34].

Prior research has used DOI partially or totally and incorporated other factors to study the acceptance of MP systems and m-wallets [4, 10, 11, 14, 34–42]. Johnson et al. (2018) used DOI by incorporating privacy risk, ubiquity, ease of use, and perceived security to study the limitations to the acceptance of m-payments services. They found that relative advantage, ease of use, perceived security, and visibility are the direct antecedents of usage intention while perceived risk, trialability, and ubiquity have indirect impacts on intention-to-use. Mombeuil (2020) used DOI to research the elements that affect the acceptance of m-wallets and noticed that the exogenous variables, namely, perceived security, perceived privacy, perceived convenience, and relative advantage have significant impacts on renewed adoption intention. Kaur et al. (2020) investigated the reasons for recommending m-wallets to others by using DOI as the base theory. They took compatibility, relative advantage, observability, complexity, and trialability as exogenous constructs while intention-to-use, intention to recommend were taken as endogenous constructs. Their results confirmed significant effects of compatibility, relative advantage, observability, and complexity on intention-to-recommend and intention-to-use while significant impacts of trialability on intention-to-recommend and intention-to-use were not found. The above studies have confirmed the appropriateness of DOI to examine the espousal of m-wallets.

Consequently, this research employs DOI as the base theory and extends it consistent with the preceding studies in the contexts of MP and m-wallets. In our proposed model along with

the DOI constructs, we consider convenience [10, 38], personal innovativeness [37, 40, 41], perceived security [10, 42], and perceived trust [35, 39, 42] as the antecedents of consumers' intention-to-use m-wallets. A detailed explanation regarding the inclusion of these variables is given in section 2.2.

## 2.2 Development of hypotheses & research model

To inspect the significant determinants of consumers' intention-to-use m-wallets, this study organised the factors into three groups namely technology characteristics (compatibility, ease-of-use, observability, trialability, convenience), behavioural beliefs (personal innovativeness, relative advantage), privacy concerns (perceived trust, perceived security). The grouping of constructs has been made based on the characteristics of the constructs. The main motivation behind such categorization is to present a comprehensive model in a simpler and easier way. Such categorization will help the practitioners to focus on the required characteristics of m-wallets. Similar practices of grouping constructs having similar characteristics were adopted by prior research [38, 42, 43]. The proposed model of this study is manifested in Fig 1. For comparison of our model with the prior literature, a review of the extant literature on the adoption of m-wallets / mobile payments is presented in Table 1 below.

**2.2.1 Technology characteristics.** *a. Compatibility (COMP).* Compatibility refers to the extent to which innovative technology is found fit to the values, needs, and experience of the user [27]. Higher values of compatibility between the users' needs and the m-wallets will lead users to enhanced intentions to use m-wallets [4]. In the contexts of m-wallets and mobile payments, researchers have found compatibility as a stronger factor of usage intention [4, 38, 41]. We believe that if consumers will find m-wallet compatible with their beliefs, needs, and experience, they will intend to use them. In the context of this study, greater compatibility means

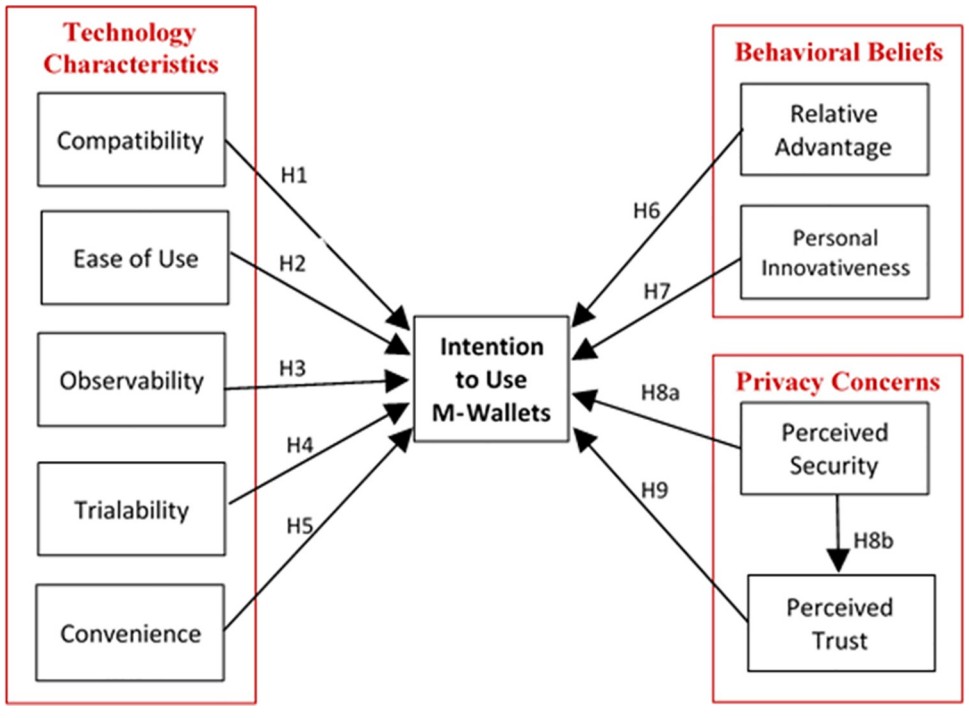

**Fig 1. Proposed model of the study.**

**Table 1. Prior literature on adoption of m-wallets / mobile payments.**

| Reference | Theory | Study Context | Constructs | Target Variable |
|---|---|---|---|---|
| [4] | DOI | M-Wallets | Relative advantage, compatibility, complexity, observability, Trialability | Intention to Use, Intention to Recommend |
| [11] | DOI, TAM | M-Payments | Ease of Use, Relative Advantage, Visibility, Perceived security, Perceived Risk, Ubiquity, Trialability | Usage Intention |
| [36] | DOI | Mobile NFC Payments | Relative advantage, compatibility, complexity, Perceived Status Benefits | Attitude Toward Innovation |
| [10] | DOI | M-Wallets | Relative Convenience, Relative advantage, Perceived security, Perceived privacy, | Renewed adoption of m-wallets |
| [38] | DOI, UTAUT | Mobile Payments | Facilitating Factors (perceived transaction convenience, compatibility, relative advantage, and social influence), Inhibiting Factors (perceived risk). Environmental factors (government support and additional value), Personal Factors (absorptive capacity, affinity, and PIIT) | Adoption Intention |
| [40] | DOI, TAM | Mobile Payments | Perceived compatibility, Subjective norms, Individual mobility, Personal innovativeness, Perceived ease of use, Perceived usefulness, Perceived security | Intention to Use |
| [41] | DOI, TAM, UTAUT | Mobile Payments | Perceived usefulness, Perceived Ease of Use, Perceived ubiquity, perceived compatibility, perceived personal innovativeness, perceived social influence, Perceived Risk, Perceived Costs | Intention to Use Mobile Payments |
| [42] | TAM, DOI | WeChat wallet | Security, Trust, Ease of use, Privacy concerns, relative advantage | Behavioural intention |
| [34] | DOI, TAM, UTAUT | Mobile Payments | Attitude towards mobile services, compatibility, Usefulness, Ease of use, Security, Intention to Use | Behavioural Intention |
| [1] | TAM, UTAUT2 | M-Wallets | Perceived Ease of Use, Usefulness, Perceived Risk, Attitude, Intention to Use, Satisfaction. Moderators: Innovativeness, Stress to Use Technology, Social Influence | Recommendation to Use |
| [44] | DOI, TAM, UTAUT | M-Wallets | Perceived Compatibility, Awareness, Perceived Usefulness, Perceived Trust, Perceived Customer Value Addition, Perceived Cost | Intention Cost |
| [35] | TAM | Mobile payment services | Relative advantage Costs, Compatibility, Ease of use, Network externalities, Trust in actors, Security, Age, Income, Use of card payments | Individual's Attitudes to Adopt the Service |
| [14] | DOI, UTAUT2 | Mobile Payments | Performance expectancy, Effort expectancy, Social influence, Facilitating conditions, Hedonic motivation, Price value, Innovativeness, Compatibility, Perceived technology security, Behavioural intention | Intention to recommend |
| [37] | DOI, TAM, UTAUT | M-Wallets | Compatibility, Perceived Ease of Use, Personal Innovativeness, Perceived Security, Social Influence, Perceived Usefulness, Rewards, Attitude | Use Intention |
| [39] | DOI, TAM | Mobile Payments | Perceived Usefulness, Perceived Ease of Use, Trust, Perceived Risk, Compatibility | Behavioural Intention |

The formation of Hypotheses development is presented in the sections below.

higher consistency between m-wallets and consumers' needs. Thus, we propose the first hypothesis:

H1. Compatibility has significant effects on intention-to-use m-wallets.

*b*. *Ease of use (EOU)*. The EOU refers to the consumers' perceptions about the simplicity and effortless functional procedure of innovation [28]. Longyara & Van (2015) has suggested that innovative technologies should be easier to use to enhance adoption. Prior research support this argument and EOU has been confirmed as the most prominent determinant of intention-to-use m-wallets/mobile payments [11, 34, 37]. Given that m-wallet is an alternative payment method to other payments methods like debit cards, credit cards, cash payment, and mobile banking, it is needed that the consumers perceive m-wallet service easier than the existing methods. If the potential users find m-wallet easier to use in making transactions conveniently, easily and quickly, their intentions to use would be influenced positively. Thus, we propose the second hypothesis:

H2. EOU has significant effects on intention-to-use m-wallets.

*c. Observability (OB)*. Observability refers to the level to which tangible results are produced by an innovation that leads to enhanced visibility [30]. Rogers (2003) posits that the more easily the consumers can view the consequences of new technology, the more likely they intend to adopt. Researchers have used observability and visibility alternatives to each other in different contexts and have found visibility as an important determinant of technology adoption [45–48]. Kaur et al. (2020) studied the espousal of m-wallets and recommending to others by consumers in India and confirmed significant impacts of observability on both intention-to-use and intention-to-recommend. Johnson et al. (2018) examined the limitations to the rapid acceptance of MP in the United States and their results showed significant effects of visibility on intention-to-use. Thus we comprehend that the visibility of m-wallets to the consumers may make them optimistic about the innovative service and a positive effect on their intention is expected. Therefore, we propose the third hypothesis:

H3. Observability has significant effects on intention-to-use m-wallets.

*d. Trialability (TR)*. Trialability means the degree to which the users can test innovation before finalising the adoption [30]. The ability of a user to try out an innovation before the adoption decision can cater more usage comfort and reduce his/her concerns about performance, usability and security [11]. According to Arvidsson (2014), adoption is an experience through which the users learn and this learning experience makes them more comfortable with the technology that leads them towards adoption. The earlier research has confirmed indirect significant effects of trialability on intention-to-use in MP/m-banking perspective and in the context of smart home technology adoption [11, 49]. In a study made by Kaur et al. (2020) in m-wallets perspective, significant impacts of trialability on intention-to-use m-wallet were not confirmed. Due to the variations in the results, it becomes important to further investigate trialability. As it is ascertained through the prior research, we expect that the consumers' ability to try out before the actual adoption will facilitate them to learn about the functionality and performance of the m-wallet service and this learning experience will boost their adoption intention. Thus, we propose the fourth hypothesis:

H4. Trialability has significant effects on intention-to-use m-wallets.

*e. Convenience (CONV)*. Convenience means the consumer's perception about the reduced time and effort needed to learn and use m-wallet in the perspective of this research. Convenience is a critical factor of consumers' acceptance of MP [16]. In comparison with other payment methods like cash and debit/credit card payments, m-wallets provide additional benefits like real-time access to finances and purchases regardless of time/space constraints [50]. M-wallets are more convenient than other payment methods because the consumers can continue browsing social media and reading news while queuing at any point-of-sale and can pay at the checkouts conveniently by simply switching to m-wallets into their smartphones [51]. Prior research has confirmed that convenience is one of the critical factors that determine the adoption of m-wallets/MP [10, 38, 52]. Keeping in view arguments of the prior research and comparing the convenience provided by m-wallets with other payment methods, we comprehend that convenience has positive impacts on consumers' intention-to-use m-wallets. Consequently, we propose the fourth hypothesis:

H5. Convenience has significant effects on intention-to-use m-wallets.

**2.2.2 Behavioural beliefs.** *a. Relative advantage (RA)*. Relative advantage describes the level to which an individual recognises an innovation better than its antecedent [27]. In simple words, an innovation's relative advantage (like m-wallet) means that the innovation is more

advantageous in comparison with the existing systems (like cash, debit/debit card payments). M-wallets provide extra benefits like convenient access and convincing features due to which the consumers may perceive m-wallets as superior service than the traditional payment methods and this perception facilitates the diffusion of the innovative service among the people [42]. Relative advantage has been studied extensively in the areas of mobile payments and m-wallets and its significant impacts on usage intention/adoption have been established [10, 11, 38, 53]. One of the main hindrances in the acceptance of m-wallets is the availability of the legacy payment methods, therefore the new payment methods (like m-wallet) must present additional benefits over the predecessors. Consequently, based on the above facts, we propose the sixth hypothesis:

H6. Relative advantage has significant effects on intention-to-use m-wallets.

*b. Personal innovativeness (PI)*. Agarwal & Prasad [54] defined personal innovativeness as the eagerness of an individual to test an innovation. The potential users having higher levels of personal innovativeness can handle higher levels of uncertainty [30] and may adopt the new technology despite having little knowledge and experience [41]. The users with more personal innovativeness have more intrinsic motivations towards the innovation and they are more likely attracted towards the information and communication technology [55]. With regard to m-wallets and mobile payments, previous studies reveal that personal innovativeness is positively related to intention to use [41, 56, 57]. Keeping in view the results of prior research, we expect positive significant influences of personal innovativeness upon intention-to-use m-wallets. Hence, we propose the seventh hypothesis:

H7. Personal Innovativeness has significant effects on intention-to-use m-wallets.

**2.2.3 Privacy concerns.** *a. Perceived security (PS)*. Perceived security refers to the users' perceptions about the security of m-wallets transactions against the risk of losing confidential information that may lead to financial losses [58]. In case of online transactions, many consumers show concerns about the fraudulent activities by hackers which can lead to the loss of important personal information [42]. Therefore, perceived security is considered one of the critical factors of new wireless technologies' adoption [40]. Thus to safeguard the electronic transactions, m-wallets service providers should warranty secure and reliable payment methods [42] otherwise security concerns may become an obstacle to the adoption of technology [56]. Empirical studies about m-wallets/MP have established significant relationship between perceived security and consumers' adoption/intention-to-use [10, 11, 14, 40]. However, few studies on NFC mobile payments did not produce such correlations [59, 60]. Thus it is reasonable to further investigate the perceived security construct in the context of m-wallets adoption. Lian [61] has posited that consumers having concerns about the security of e-payments will have a trust deficit in these services. Prior research has confirmed Significant impacts of perceived security on trust in the contexts of smartphone banking and e-commerce [62, 63]. Hence, we postulate the eighth hypothesis:

H8. Perceived security has significant effects on (a) intention-to-use m-wallets (b) perceived trust.

*b. Perceived trust (PT)*. Perceived trust refers to the physiological state of the consumers to accept risks in online transactions based upon their positive expectations from the behaviours and intentions of the service provider [42]. The significance of trust in examining m-wallets is natural because financial matters and payments are totally based on trust [35]. Consumers' trust in technologies positively influence their intention to use the technology since they

perceive it safe, reliable, and trustworthy and they decide to use the technology [64]. In case of online transactions, the consumers are unable to test or ensure security measures when putting their personal information before carrying out transactions is a challenging task [42]. Consumers exercise repurchase behaviours when they have trust in the retailer [65]. Thus we understand that perceived trust is vital for the adoption of m-wallets because the consumers need to submit their confidential information when they want to use it for the first time. Therefore, this study expects significant impacts of perceived trust on intention-to-use. Prior research provides support to this argument [42, 44, 64, 66]. Hence, we postulate the ninth following hypothesis:

H9. Perceived trust has significant impacts on intention-to-use m-wallets.

**2.2.4 Conceptual model.** The proposed model of the study is depicted in Fig 1 above.

# 3. Research methodology

## 3.1 Instrument development

Measurement items of the proposed model constructs were acquired from the prior relevant literature and adapted to the m-wallets context to ensure construct validity of measurement scales. Measurement items for compatibility, relative advantage, observability, trialability, and intention-to-use were adopted from [4]. Items for perceived security and perceived trust were derived from [42], items for convenience were taken from [38], items for ease of use were acquired from [40], and items for personal innovativeness were drawn from [41]. Relative advantage, ease of use, compatibility, convenience, perceived trust, and perceived security were estimated utilizing four items each. Observability and trialability were measured by using two items each, personal innovativeness used three items while the intention-to-use m-wallets was measured by using five items. Measurement items of the study are presented as S1 Text.

The original questionnaire was written in English but it was intended to serve m-wallet users in Saudi Arabia. Therefore, it was translated into Arabic by a professional native translator. To ensure consistency between the English and Arabic versions, the questionnaire was translated back into English. A definite version of the Arabic version was prepared after analysing the differences. English and Arabic versions of questionnaire are listed in S2 and S3 Texts.

## 3.2 Data collection & sample

To collect the data for the survey, a convenience sampling method was employed since the exact number of m-wallets users is now not known for this study. For this purpose, both online and hardcopies of questionnaires were administered and quota sampling was used to get matching with the age and gender characteristics of the target population [67]. Hardcopies data collection was carried out in five main cities of Saudi Arabia namely Riyadh, Jeddah, Dammam, Madinah and Taif. A screening question "Do you use any Mobile Wallet Application?" was added at the beginning of the questionnaire to ensure responses from m-wallets users. We received 783 responses. During the data screening stage, 46 responses were rejected due to missing data. In remaining 737 valid responses, 51.7% were male and 48.3% were female. Age-wise, 29.2% respondents were 16–25 years old, 25.9% were 26–35 years old, 23.5% were in the range 36–45 years, 16.3% were in the age 46–55 years, and 5.2% were above 55 years. Complete dataset is provided as S1 Dataset.

**Table 2. ANOVA table.**

| Relationships | | Sum of Squares | df | Mean Square | F | Sig. | Linear |
|---|---|---|---|---|---|---|---|
| IU * COMP | Deviation from Linearity | 151.232 | 178 | 0.850 | 1.827 | 0.000 | No |
| IU * CONV | Deviation from Linearity | 152.754 | 150 | 1.018 | 1.941 | 0.000 | No |
| IU * EOU | Deviation from Linearity | 51.714 | 71 | 0.728 | 1.051 | 0.370 | Yes |
| IU * OB | Deviation from Linearity | 35.433 | 23 | 1.11 | 2.368 | 0.000 | No |
| IU * PI | Deviation from Linearity | 56.663 | 71 | 0.798 | 1.213 | 0.121 | Yes |
| IU * PS | Deviation from Linearity | 173.346 | 98 | 1.769 | 3.214 | 0.000 | No |
| IU * PT | Deviation from Linearity | 163.295 | 99 | 1.649 | 2.850 | 0.000 | No |
| IU * RA | Deviation from Linearity | 106.724 | 156 | 0.684 | 1.147 | 0.133 | Yes |
| IU * TR | Deviation from Linearity | 14.353 | 19 | 0.755 | 1.231 | 0.233 | Yes |

**Note:** IU: Intention-to-use; COMP: Compatibility; EOU: Ease of Use; CONV: Convenience; OB: Observability; PI: Personal Innovativeness; PS: Perceived Security; PT: Perceived Trust; RA: Relative Advantage; TR: Trialability.

# 4. Statistical analysis and results

To assess our proposed model, we used a twofold analysis. At stage one, PLS-SEM was employed to examine the significance of the hypothesised paths. Artificial neural network (ANN) analysis was used at stage two to endorse the outcomes of PLS-SEM and to measure the relative importance of the SEM-based significant determinants. According to Urbach & Ahlemann [68], the PLS-SEM is more appropriate for a model containing more constructs. Before conducting the multivariate analysis, assessment of the multivariate practices (like normal distribution, linearity, and multicollinearity) is important [69]. To test the normality of the data, we used the one-sample Kolmogorov-Smirnov test and we found that data distribution is non-normal. It further supported us for the selection of partial least squares (PLS) due to its robustness against the non-normal distribution of data [70]. To assess the linearity of relationships, an ANOVA test was used. It is evident from the results in Table 2 that linear, as well as non-linear relationships, are contained in the model. To examine the multicollinearity issue, the VIF and tolerance values were evaluated. The VIF values are found between 1.714 to 3.138, which is less than the threshold 10 and the tolerance values range between 0.319 to 0.583 which is larger than 0.10 showing that the multicollinearity issue is not prevalent in our data [7].

The predictors and dependent were measured using the same scale, therefore we examined the existence of common method bias (CMB) by using Harman's single factor test. Results of this test indicate that a single factor extracted 32.18% of the variance. Since it is far less than 50%, so we conclude that CMB is not an issue for our data. To get further confirmation, a full collinearity test was also conducted and we found all variance inflation factor (VIF) values are below 3.3. According to Kock [71], VIF values below 3.3 indicate the absence of CMB in the data.

Since the data distribution is non-normal, therefore the partial least squares structural equation modelling (PLS-SEM) is a more appropriate method than factor-based SEM [72]. Applying the twofold analysis in which the ANN follows the PLS-SEM is well suited due to the presence of non-linear relationships as the factor-based SEM and composite-based SEM are unable to treat non-linear relationships [73].

In many situations, linear statistical methods like multiple regression analysis and SEM are not adequate to model the complicated processes of human decision making [74]. These methods normally oversimplify the complexities of the acceptance decisions due to their capability of assessing linear models only [75]. To deal with this type of problem, the ANN technique is

recommended which can identify linear and non-linear relationships [76]. The ANN method does not need the fulfilment of any distribution assumption like linearity, normality, or homoscedasticity [7, 75]. Moreover, the ANN models are substantially robust and their prediction accuracy is more than the linear models [40, 75]. The ANN structure is based on human brain architecture in which neurons are similar to biological neurons in the human brain [40]. The ANN technique uses its learning process to obtain knowledge [77]. Due to the learning ability of ANN, it is distinguished as a preferable method over other statistical methods [78]. Without depending on a theoretical model, the ANN uses artificial neurons to link the input and output layers and their interrelationships in a hidden layer to enhance the prediction power [69]. Since the ANN is using the black box operation due to which it cannot evaluate the significance level of the inter-node relationships, therefore this technique is inappropriate for hypotheses testing [76]. Thus using a twofold SEM–ANN analysis would complement one another as the PLS-SEM is appropriate to evaluate the linear relationships but it cannot evaluate the non-linear relationships while the ANN can identify non-linear relationships but it is inappropriate to test the hypotheses [7]. For this purpose, first, we used the PLS-SEM to measure the effects of independent variables on the dependent variable, while in stage two we employed the ANN analysis to inspect the relative importance of the significant determinants towards endogenous variables [40, 73].

## 4.1 PLS-SEM analysis

**4.1.1 Assessment of measurement model.** To evaluate the constructs' reliability and validity, the PLS algorithm was employed with standard settings. Table 3 lists the results of reliability and convergent validity. It is obvious from these results that values of Cronbach's alpha, composite reliability, and indicators' reliability are above 0.7 showing a high degree of the measurement model's reliability [7]. Two indicators EOU4 (0.384) and IU5 (0.603) were removed due to outer loading less than 0.7. Moreover, the average variance extracted (AVE) values are more than 0.5 which confirms the convergent validity of the scales [79].

To examine the discriminant validity, we assessed the Fornell-Larcker criterion and Heterotrait-Monotrait ratio (HTMT) criterion. Results of these tests are exhibited in Table 4. The diagonal elements show the AVE's square roots of different constructs which are higher than their corresponding correlations with other variables [7]. It reveals that discriminant validity is established. The HTMT is listed above the diagonal elements. All the HTMT ratios are below 0.9 which further confirms the existence of discriminant validity [80].

**4.1.2 Structural model analysis.** For hypotheses testing, the bootstrapping process was used in SmartPLS 3.3 with 5000 subsamples, Bias-Corrected and Accelerated (BCA) Bootstrap, and a two-tailed test with a significance level of 0.05. To evaluate the significance of the relationships, the path coefficients with the related t and p values were assessed. Bootstrapping results are covered in Table 5. Path analysis in Fig 2 shows that all the hypothesised relationships except the TR → IU path are significant at a significant level of at least $p < 0.05$.

These results indicate that the technology characteristics like compatibility (β: 0.243, $p < 0.01$), ease of use (β: 0.115, $p < 0.01$), observability (β: 0.134, $p < 0.01$), and convenience (β: 0.169, $p < 0.05$) have significant impacts on intention-to-use m-wallets thus confirming our hypotheses H1, H2, H3, H5. The strengths of relationships of compatibility, observability and convenience with intention-to-use are comparatively stronger than the strength of ease of use with the dependent variable. The effects of trialability (β: 0.027, $p > 0.1$) on intention-to-use m-wallets are insignificant, thus rejecting our hypothesis H4. These findings indicate that the consumers consider compatibility, ease of use, observability and convenience more important for the adoption of m-wallets and trialability is not a critical factor. This illustrates that after

**Table 3. Reliability & convergent validity tests summary.**

| Construct | α >0.7 | Composite Reliability >0.7 | Items | Indicators' reliability > = 0.7 | AVE >0.5 |
|---|---|---|---|---|---|
| Compatibility | 0.847 | 0.897 | COMP1 | 0.858 | 0.686 |
| | | | COMP2 | 0.857 | |
| | | | COMP3 | 0.845 | |
| | | | COMP4 | 0.750 | |
| Convenience | 0.845 | 0.896 | CONV1 | 0.843 | 0.684 |
| | | | CONV2 | 0.751 | |
| | | | CONV3 | 0.854 | |
| | | | CONV4 | 0.856 | |
| Ease of Use | 0.836 | 0.902 | EOU1 | 0.870 | 0.754 |
| | | | EOU2 | 0.866 | |
| | | | EOU3 | 0.801 | |
| | | | **EOU4** | **0.384** | |
| Intention to Use | 0.879 | 0.917 | IU1 | 0.829 | 0.734 |
| | | | IU2 | 0.837 | |
| | | | IU3 | 0.851 | |
| | | | IU4 | 0.865 | |
| | | | **IU5** | **0.603** | |
| Observability | 0.755 | 0.891 | OB1 | 0.900 | 0.803 |
| | | | OB2 | 0.892 | |
| Personal Innovativeness | 0.836 | 0.902 | PI1 | 0.895 | 0.754 |
| | | | PI2 | 0.891 | |
| | | | PI3 | 0.817 | |
| Perceived Security | 0.877 | 0.916 | PS1 | 0.866 | 0.731 |
| | | | PS2 | 0.829 | |
| | | | PS3 | 0.852 | |
| | | | PS4 | 0.871 | |
| Perceived Trust | 0.872 | 0.913 | PT1 | 0.858 | 0.723 |
| | | | PT2 | 0.842 | |
| | | | PT3 | 0.842 | |
| | | | PT4 | 0.859 | |
| Relative Advantage | 0.751 | 0.843 | RA1 | 0.791 | 0.573 |
| | | | RA2 | 0.769 | |
| | | | RA3 | 0.725 | |
| | | | RA4 | 0.740 | |
| Trialability | 0.817 | 0.916 | TR1 | 0.920 | 0.845 |
| | | | TR2 | 0.919 | |

finding m-wallet service easy to use, convenient, observable, and compatible with their lifestyle and experience, they intend to use it and are not interested in further trialability. The impacts of the behavioural beliefs namely relative advantage (β: 0.164, p<0.05), and personal innovativeness (β: 0.140, p<0.05) on intention-to-use are significant. These findings support our hypotheses H6, and H7. These outcomes reveal that consumers' innovativeness towards innovations and their perceptions that the innovation is more advantageous are playing critical functions in affecting their intention to use. Our findings also confirm the significant effects of privacy concerns over the consumers' intention-to-use m-wallets i.e. perceived security (β: 0.128, p<0.01), and perceived trust (β: 0.108, p<0.01). Moreover, significant impacts of

**Table 4. Discriminant validity.**

|         | COMP  | CONV  | EOU   | IU    | OB    | PI    | PS    | PT    | RA    | TR    |
|---------|-------|-------|-------|-------|-------|-------|-------|-------|-------|-------|
| **COMP** | **0.829** | 0.461 | 0.538 | 0.766 | 0.502 | 0.641 | 0.464 | 0.482 | 0.545 | 0.229 |
| **CONV** | 0.394 | **0.827** | 0.537 | 0.704 | 0.545 | 0.482 | 0.428 | 0.462 | 0.635 | 0.268 |
| **EOU**  | 0.456 | 0.454 | **0.868** | 0.643 | 0.438 | 0.477 | 0.338 | 0.403 | 0.522 | 0.249 |
| **IU**   | 0.666 | 0.614 | 0.553 | **0.857** | 0.696 | 0.669 | 0.611 | 0.599 | 0.763 | 0.340 |
| **OB**   | 0.404 | 0.438 | 0.348 | 0.569 | **0.896** | 0.418 | 0.456 | 0.461 | 0.663 | 0.283 |
| **PI**   | 0.543 | 0.407 | 0.399 | 0.575 | 0.332 | **0.868** | 0.391 | 0.357 | 0.486 | 0.211 |
| **PS**   | 0.403 | 0.372 | 0.290 | 0.538 | 0.371 | 0.334 | **0.855** | 0.483 | 0.491 | 0.352 |
| **PT**   | 0.414 | 0.400 | 0.346 | 0.528 | 0.374 | 0.307 | 0.423 | **0.850** | 0.445 | 0.255 |
| **RA**   | 0.438 | 0.510 | 0.413 | 0.622 | 0.500 | 0.385 | 0.398 | 0.361 | **0.757** | 0.316 |
| **TR**   | 0.192 | 0.223 | 0.206 | 0.291 | 0.223 | 0.175 | 0.298 | 0.216 | 0.247 | **0.919** |

Notes: 1. The diagonal elements express the square root of the AVE. The elements above the diagonal are HTMT ratios. While elements below the diagonal elements are the correlations between the constructs.

2. COMP: Compatibility; CONV: Convenience; EOU: Ease of Use; IU: Intention-to-use; OB: Observability; PI: Personal Innovativeness; PS: Perceived Security; PT: Perceived Trust; RA: Relative Advantage; TR: Trialability.

perceived security (β: 0.423, p<0.01) on perceived trust were also established. Hence, our hypotheses H8a, H8b and H9 are supported. Conclusively, the empirical findings validated our conceptual model which is capable to explain 72.2% of the variance in intention-to-use m-wallets.

**4.1.3 Importance–Performance Map Analysis (IPMA).** To evaluate the importance and performance of the latent variables, we employed IPMA in SmartPLS 3.3. The IPMA is helpful in identifying such latent variables that have comparatively higher importance but relatively lower performance to predict the target construct. To analyse the IPMA, it is divided into four parts. The lower-right (higher importance, lower performance) is the area that provides the opportunity for the highest improvement. The order of consideration of remaining areas for improvement is upper-right, lower-left, and upper-left. The IPMA results are shown in Fig 3. The performance is shown on the vertical axis while the horizontal axis shows importance. The mean values of importance and performance are 0.123 and 70.38 respectively. The

**Table 5. Summary of structural model path coefficients.**

| Hyp. # | Path | Path Coefficient | Standard Deviation | T Statistics | P Values | Sig. Level | Remarks |
|--------|------|------------------|--------------------|--------------|----------|------------|---------|
| H1 | COMP → IU | 0.243 | 0.028 | 8.659 | 0.000 | *** | Supported |
| H2 | EOU → IU | 0.115 | 0.029 | 4.014 | 0.000 | *** | Supported |
| H3 | OB → IU | 0.134 | 0.027 | 4.919 | 0.000 | *** | Supported |
| H4 | TR → IU | 0.027 | 0.021 | 1.253 | 0.211 | **N.S.** | Not Supported |
| H5 | CONV → IU | 0.169 | 0.026 | 6.623 | 0.000 | *** | Supported |
| H6 | RA → IU | 0.164 | 0.028 | 5.931 | 0.000 | *** | Supported |
| H7 | PI → IU | 0.140 | 0.025 | 5.649 | 0.000 | *** | Supported |
| H8a | PS → IU | 0.128 | 0.027 | 4.757 | 0.000 | *** | Supported |
| H8b | PS → PT | 0.423 | 0.035 | 12.022 | 0.000 | *** | Supported |
| H9 | PT → IU | 0.108 | 0.027 | 3.974 | 0.000 | *** | Supported |

Notes: 1. $^*p < 0.1$; $^{**}p < 0.05$; $^{***}p < 0.01$; NS = Not Significant.

2. COMP: Compatibility; CONV: Convenience; EOU: Ease of Use; IU: Intention-to-use; OB: Observability; PI: Personal Innovativeness; PS: Perceived Security; PT: Perceived Trust; RA: Relative Advantage; TR: Trialability.

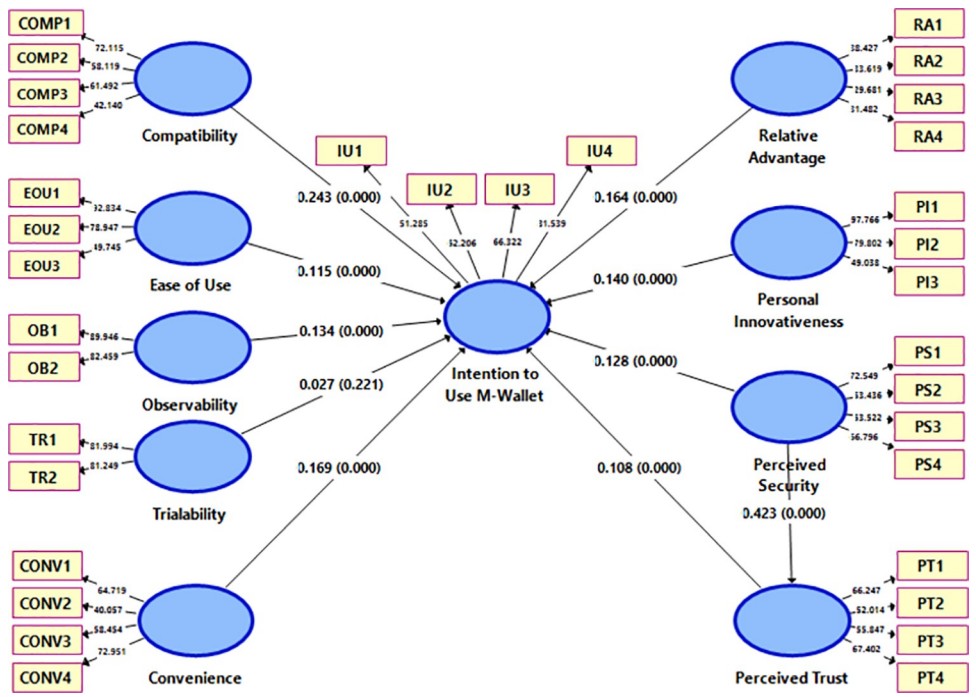

**Fig 2. SEM analysis of conceptual model.**

perceived trust construct exhibits the highest performance 74.8%. The next five constructs whose performance is higher than the mean are perceived security (73.31%), observability (72.66%), personal innovativeness (72.4%), ease of use (72.39%), and relative advantage (71.99%). The performance values for the remaining constructs are trialability (67.14%),

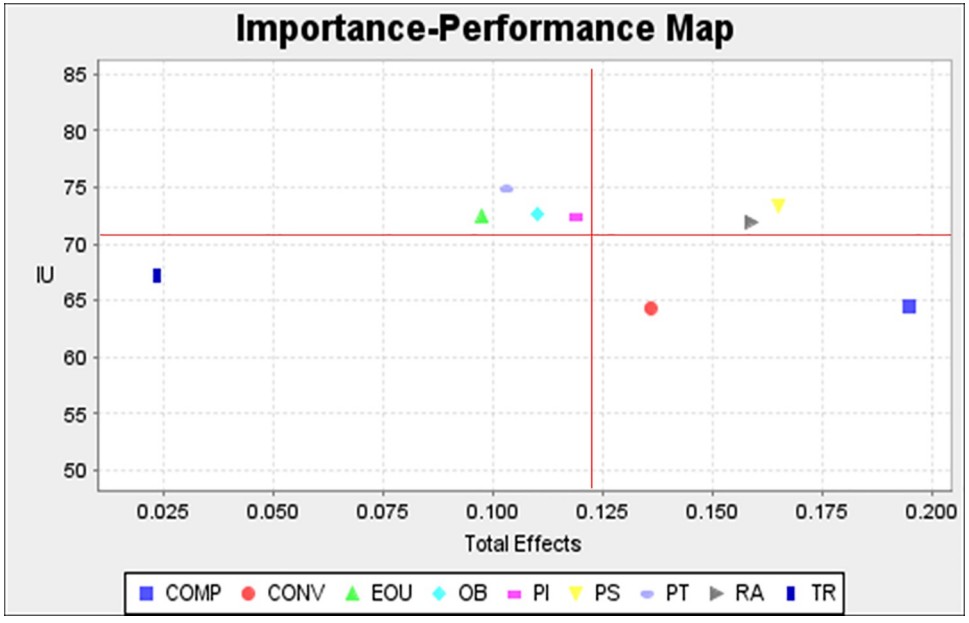

**Fig 3. Importance–Performance Map Analysis.**

compatibility (64.42%), and convenience (64.33%). It is evident that compatibility and convenience are located in the lower-right area whose importance is higher than the mean and there is more potential to improve their performance. The next two variables that have the potential for improvement are perceived security and relative advantage. In the next stage, the managers should focus on the following areas: ease of use, perceived trust, and observability. The performance of personal innovativeness is above average and their importance is very near to the mean. There is less chance of further improvement in this construct.

## 4.2 ANN analysis

This study used SPSS 23 to perform multilayer perception ANN analysis. The ANN method has been described at length by researchers as to how the neurons acquire knowledge through a learning process and use this knowledge to predict the output neuron nodes [70, 81]. This study included only those constructs for ANN analysis whose statistical significance was confirmed by PLS-SEM [40]. Hence, the trialability construct was not included in the ANN model as shown in Fig 4. Following the practice used by many researchers in most of the technology adoption neural network models [7, 40, 81], we employed one hidden layer in this study as one hidden layer is enough to portray any continuous function [82]. The generation of hidden layers was set to automatic and the sigmoid activation function was used for hidden and output layers [81]. To avoid over-fitting issues, a ten-fold cross-validation procedure was employed with 90% data allocation for training and 10% for testing of the networks [83].

To measure the ANN model's predictive accuracy, we obtained the root mean square of errors (RMSE) values for the testing and training stage in each round. Values in Table 6 show that the average RMSE values for testing and training stages are 0.078 and 0.075 respectively.

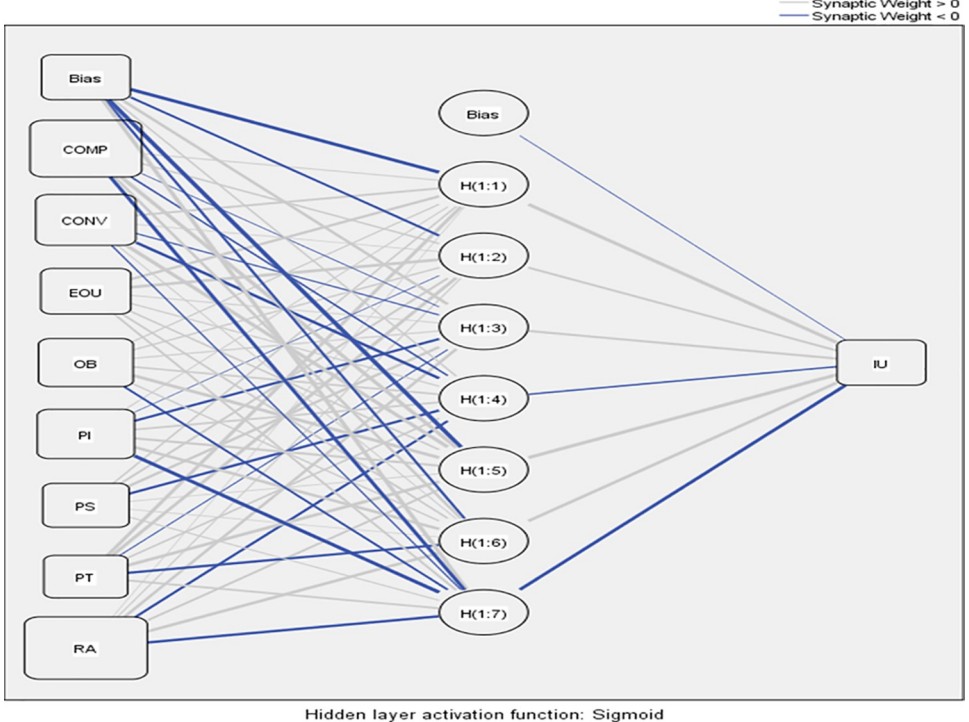

Hidden layer activation function: Sigmoid
Output layer activation function: Sigmoid

**Fig 4. The ANN model.**

**Table 6. RMSE values during training and testing stages.**

| Neural Networks | ANN Model ($R^2$ = 83.2%) | | | | | |
|---|---|---|---|---|---|---|
| | Training | | | Testing | | |
| | N1 | SSE | RMSE | N2 | SSE | RMSE |
| ANN1 | 662 | 3.821 | 0.076 | 75 | 0.354 | 0.069 |
| ANN2 | 659 | 3.778 | 0.076 | 78 | 0.401 | 0.072 |
| ANN3 | 655 | 3.446 | 0.073 | 82 | 0.545 | 0.082 |
| ANN4 | 650 | 3.632 | 0.075 | 87 | 0.507 | 0.076 |
| ANN5 | 659 | 3.835 | 0.076 | 78 | 0.356 | 0.068 |
| ANN6 | 665 | 3.549 | 0.073 | 72 | 0.418 | 0.076 |
| ANN7 | 660 | 4.795 | 0.085 | 77 | 0.399 | 0.072 |
| ANN8 | 661 | 4.152 | 0.079 | 76 | 0.463 | 0.078 |
| ANN9 | 658 | 3.949 | 0.077 | 79 | 0.457 | 0.076 |
| ANN10 | 656 | 5.136 | 0.088 | 81 | 0.573 | 0.084 |
| Average | | 4.009 | 0.078 | | 0.447 | 0.075 |
| St Dev | | 0.548 | 0.005 | | 0.076 | 0.005 |

Notes

1. N = number of samples, SSE = Sum of squares errors, RMSE = root mean square of errors.

2. In the ANN Model, Compatibility; Ease of Use; Convenience; Observability; Personal Innovativeness; Perceived Security; Perceived Trust; Relative Advantage; and Trialability served as the input neurons; while Intention-to-use served as the output neuron.

3. $R^2$ = 1—RMSE/$S^2$, where $S^2$ is the variance of the desired output for the test data.

These values are reasonably small which confirms the predictive accuracy of the ANN model [7, 83]. To examine the performance of the ANN model, we evaluated the percentage of variance ($R^2$) described by the ANN model applying the formula $R^2 = 1 - \frac{RMSE}{S^2}$ where $S^2$ is the variance of preferred-output [84]. Our findings of $R^2$ reveal that the input neurons are capable to predict 83.2% of the variance in the Intention-to-use m-wallets. The $R^2$ (83.2%) value obtained from the ANN is higher than the $R^2$ (72.2%) value obtained from SEM analysis, which indicates that the ANN model has a higher predictive capability of 83.2% to describe m-wallets adoption.

To determine the strengths of the predictive capability of the input neurons, we carried out a sensitivity analysis. By employing the sensitivity analysis, the relative importance is acquired to rank the exogenous constructs [69]. For each determinant, the relative importance is the degree of the change occurring in the magnitude of the predicted output with the various measures of the determinant [85]. The relative importance of each construct is used to evaluate its normalised importance by dividing its average relative importance by the highest relative importance and expressing the result in percentage form [7, 40]. Results depicted in Table 7 indicate that perceived security (86.4%) is the prime determinant of consumers' intention-to-use m-wallets. The compatibility and observability constructs come at the next level of higher importance which have normalised importance values 82.9% and 81.5% respectively. These are followed by convenience (78.5%), relative advantage (77.2%), personal innovativeness (74%), perceived trust (63.3%), and ease of use (62.5%).

## 5. Discussions and conclusions

This research aims to study the key determinants of consumers' intentions-to-use m-wallets. To accomplish this objective, a model was constituted based on DOI by integrating compatibility, observability, ease of use, trialability, convenience, relative advantage, perceived security,

Table 7. Sensitivity analysis with normalized importance.

| Neural Network | ANN Model (Output Neuron: Intention-to-use) | | | | | | | |
|---|---|---|---|---|---|---|---|---|
| | COMP | EOU | CONV | OB | PI | PS | PT | RA |
| ANN1 | 0.159 | 0.111 | 0.143 | 0.130 | 0.107 | 0.104 | 0.072 | 0.174 |
| ANN2 | 0.163 | 0.130 | 0.101 | 0.114 | 0.122 | 0.091 | 0.084 | 0.195 |
| ANN3 | 0.149 | 0.113 | 0.128 | 0.113 | 0.128 | 0.118 | 0.087 | 0.165 |
| ANN4 | 0.145 | 0.111 | 0.125 | 0.107 | 0.118 | 0.115 | 0.079 | 0.200 |
| ANN5 | 0.168 | 0.133 | 0.103 | 0.109 | 0.110 | 0.098 | 0.080 | 0.199 |
| ANN6 | 0.145 | 0.125 | 0.133 | 0.108 | 0.141 | 0.093 | 0.091 | 0.164 |
| ANN7 | 0.124 | 0.204 | 0.080 | 0.093 | 0.177 | 0.114 | 0.122 | 0.087 |
| ANN8 | 0.184 | 0.118 | 0.088 | 0.119 | 0.117 | 0.087 | 0.045 | 0.241 |
| ANN9 | 0.144 | 0.135 | 0.101 | 0.087 | 0.156 | 0.091 | 0.057 | 0.229 |
| ANN10 | 0.145 | 0.094 | 0.120 | 0.137 | 0.134 | 0.109 | 0.055 | 0.206 |
| Average relative importance | 0.153 | 0.127 | 0.112 | 0.112 | 0.131 | 0.102 | 0.077 | 0.186 |
| Normalized relative importance (%) | 82.9% | 62.5% | 78.5% | 81.5% | 74.0% | 86.4% | 63.3% | 77.2% |

Note: COMP: Compatibility; EOU: Ease of Use; CONV: Convenience; OB: Observability; PI: Personal Innovativeness; PS: Perceived Security; PT: Perceived Trust; RA: Relative Advantage; TR: Trialability.

personal innovativeness, and perceived trust. This study used a twofold approach comprising of SEM-ANN analysis to analyse the propositioned model. The PLS-SEM technique helped in assessing the hypotheses and the ANN approach supported in validating the outcomes of the PLS-SEM. Thus, this research encourages the application of a multi-analytical methodology to treat linear as well as non-linear relationships, and improve the model' prediction precision.

Our results confirm the significant consequences of compatibility on intention-to-use m-wallets. These outcomes have consistency with the discoveries established by [4, 38, 41]. These outcomes suggest that if individuals find m-wallets fit their needs, lifestyle, and experience, it will affect their intention to use. The ease-of-use construct was also observed as a critical antecedent of intention-to-use. These results are in accordance with the findings given by [11, 34, 37]. It implies that the individuals' intention-to-use m-wallet service is determined by the ease of use factor and they dislike a complicated or difficult-to-use service. Our hypothesis H3 is about the noteworthy effects of observability on intention-to-use and our findings supported it. These outcomes hold up the findings of [4]. The consumers observe others at superstores and other retail outlets using m-wallets and they are attracted towards its use. Significant effects of convenience are on intention-to-use were also confirmed which provides support for prior research [10, 38, 52]. The use of M-wallets is a voluntary action and the users will use it if they feel more convenient in utilising it. We found significant impacts of relative advantage on intention-to-use. These results are identical to the findings revealed by [10, 11, 38, 53]. It shows that the individuals intend to use m-wallets if they perceive m-wallets as more advantageous in comparison with the conventional payment methods. We also found significant effects of personal innovativeness on intention-to-use and these results endorse the outcomes given by [41, 56, 57]. It reveals that the individuals with a higher degree of personal innovativeness will be the early adopters of m-wallets. Additionally, we found significant impacts of perceived trust and perceived security on intention-to-use. Our findings about the significant influences of perceived security are in line with the outcomes presented by [10, 11, 14, 40] while the positive significant impacts of perceived trust uphold the findings of [42, 44, 64]. Moreover, the ANN outcomes reveal that the perceived security has been confirmed as the most prominent predictor of consumers' intention-to-use m-wallets. Our results

demonstrated insignificant effects of trialability on intention-to-use m-wallets. This finding supported the results demonstrated by [4]. This research has collected data from those respondents who have prior experience of using m-wallets. This can be a possible reason for the weak relationship between trialability and intention as those respondents have crossed the stage of trialability and due to higher degrees of personal innovativeness, they have adopted m-wallets already and they do not consider a trail use necessary for the actual adoption.

## 5.1 Theoretical and practical implications

This research delivers important theoretical contributions to the body of knowledge relevant to the technology innovations, and specifically to m-wallet literature. The study exhibits that the DOI is an effective theory to investigate the determinants of m-wallets adoption. Numerous research papers have concentrated on the espousal of m-wallets but few scholars have considered DOI as the base theory in this context. This research highlights the significance of DOI by incorporating other important constructs like personal innovativeness, convenience, perceived security, and perceived trust to explore the transition of consumers from conventional payments methods to NFC based payment methods like m-wallet.

We have successfully presented a validated model based on the DOI by combining it with technological features, behavioural features, privacy concerns and our model is capable to provide comprehensive insights on the influential factors of consumers' intention-to-use m-wallets. SEM-based results of our study present a parsimonious model with the capability of explaining 72.2% of the variance in the intention-to-use m-wallets. The ANN model complemented this finding by providing $R^2$ = 83.2%.

Using a twofold SEM–ANN approach, this study contributes to the literature on m-wallets by providing new insights from assessing the SEM and ANN models and capturing deeper insights about the espousal of m-wallets. The SEM–ANN approach helped us in capturing the linear and non-linear relationships between dependent and independent variables. This research used the SEM method to ascertain the strengths of the hypothesised paths while the ANN approach was employed to find the relative importance of exogenous constructs towards endogenous construct. Thus, our study offers contributions to the extant literature by providing prospects to compare and scrutinise the predictive accuracy of both techniques.

This research has tested SEM-based IPMA and ANN-based sensitivity analysis and our findings indicate that perceived security exhibits the highest performance. This result is unique in the context of m-wallets adoption as different results were given by other researchers. For example, Johnson et al. (2018) found the ease of use construct as the principal determining factor while Liébana-Cabanillas et al. (2018) found compatibility as the most important determinant of m-wallet adoption. Other results of our research are mostly in accordance with the extant research.

Some similarities and variances were also found in the findings of SEM and ANN models. For instance, with regard to the determinants' importance, similar results were produced by both SEM and ANN models keeping perceived trust at the seventh position. Slight differences were found in the importance of constructs compatibility, observability, convenience, and personal innovativeness. SEM analysis kept compatibility at the second position while ANN analysis kept it at the first position. Observability appears as the third important construct according to SEM results while it is at fourth important construct in ANN analysis. Similarly, convenience stands at fourth while ANN shows it as the second important construct. The SEM and ANN findings show personal innovativeness at sixth and fifth positions respectively. SEM positioned perceived security at first while ANN positioned it at the sixth position. The relative advantage was ranked at the fifth position by SEM while the ANN ranked it at the eighth

position. The ease of use construct was ranked by SEM and ANN at eighth and third positions respectively. Such results open doors for future scholars to investigate further about the phenomenon.

In terms of practical implications, the current study provides important insights for practitioners and retailer marketers. First, the findings denote that the technology features like compatibility, ease of use, observability, and convenience affect the consumers' intentions-to-use m-wallets. It suggests that practitioners should ensure the availability of these features in m-wallet service.

Second, behavioural features like personal innovativeness and relative advantage have important effects on the espousal of m-wallets. Therefore, the practitioners should concentrate on equipping m-wallets with more value-added services so that m-wallets become more advantageous. Moreover, the companies can concentrate on launching special customer-centric campaigns so as they get awareness about the convenience and benefits related to the conventional payment methods.

Third, the current study demonstrates that perceived security has significant effects on consumers' adoption of m-wallets. These findings suggest that the companies should concentrate on providing complete security to m-wallet transactions. The developers should give prime importance to the security features of m-wallets. The retailing companies should collaborate with cyber-security companies to secure transactions through m-wallets. In this regard, multi-level security measures like OTP (one-time password), fingerprint authentication, and sophisticated encryption methods can enhance the consumers' perceptions about the availability of security.

Fourth, this study finds that perceived trust has the highest importance within the determinants of consumers' intention-to-use m-wallets. It suggests the companies should take every possible step to enhance security, minimise the risks inherent in fiscal transactions and thus increase the consumers' trust and confidence in the security system of m-wallets. A higher level of consumers' trust will boost the adoption of m-wallets.

## 5.2 Limitations and future research avenues

Despite the important contributions, some limitations are associated with this study which open doors for future research avenues. First, the study presents a validated model in the context of m-wallets adoption with a reasonable predictive capability (SEM-based model: $R^2 =$ 72.2%, and ANN-based model: $R^2 = 83.2\%$). Future studies can test the model of this study in other contexts of information systems. Second, this research used a cross-sectional method to collect respondents' feedback at a one-time span. To cover the temporal effects, future studies may employ a longitudinal approach to examine the consumers' behaviours. Third, this research has tested a model to investigate the determining factors of m-wallets adoption that is a particular NFC MP method. Scholars can conduct future studies from the perspectives of other NFC mobile payments. Fourth, our study used sample data from Saudi Arabia. Future research may conduct cross-cultural or cross-nation studies to extend the scope of this study. Fifth, we have not considered moderating effects of any variable in our study. It is suggested that future investigations should incorporate moderating effects of age, gender, educational level, and income level, which can bring to fore valuable and interesting results. Finally, this research has covered the sample from Saudi Arabia only. Future studies may consider cross-cultural analysis as the availability of facilitating conditions may affect the users' espousal of M-wallets.

## Supporting information

**S1 Text. Measurement items.**
(DOCX)

**S2 Text. Survey questionnaire–English.**
(DOCX)

**S3 Text. Survey questionnaire–Arabic.**
(DOCX)

**S1 Dataset. Mobile wallets adoption in KSA.**
(CSV)

## Acknowledgments

The authors are thankful to the Deanship of Scientific Research at King Saud University represented by the Research Centre in the College of Business Administration for supporting this research.

## Author Contributions

**Conceptualization:** Imdadullah Hidayat-ur-Rehman, Saeed Alzahrani, Mohd Ziaur Rehman, Fahim Akhter.

**Data curation:** Imdadullah Hidayat-ur-Rehman.

**Formal analysis:** Imdadullah Hidayat-ur-Rehman.

**Methodology:** Imdadullah Hidayat-ur-Rehman.

**Writing – original draft:** Imdadullah Hidayat-ur-Rehman, Saeed Alzahrani.

**Writing – review & editing:** Mohd Ziaur Rehman, Fahim Akhter.

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
