## [Decision Letter · Decision Letter 0]

24 Nov 2021

PONE-D-21-29301Determining the factors of M-Wallets adoption. A twofold SEM-ANN approachPLOS ONE

Dear Dr. Hidayat-ur-Rehman,

Thank you for submitting your manuscript to PLOS ONE. After careful consideration, we feel that it has merit but does not fully meet PLOS ONE’s publication criteria as it currently stands. Therefore, we invite you to submit a revised version of the manuscript that addresses the points raised during the review process.

The paper needs a MAJOR REVISION. Authors should address what the reviewers highlighted in order to improve the readability of the paper. Please submit your revised manuscript by Jan 08 2022 11:59PM. If you will need more time than this to complete your revisions, please reply to this message or contact the journal office at plosone@plos.org. Please include the following items when submitting your revised manuscript:A rebuttal letter that responds to each point raised by the academic editor and reviewer(s). You should upload this letter as a separate file labeled 'Response to Reviewers'.A marked-up copy of your manuscript that highlights changes made to the original version. You should upload this as a separate file labeled 'Revised Manuscript with Track Changes'.An unmarked version of your revised paper without tracked changes. You should upload this as a separate file labeled 'Manuscript'.

We look forward to receiving your revised manuscript.

Kind regards,

Barbara Guidi

Academic Editor

PLOS ONE

Journal Requirements:

Reviewers' comments:

Reviewer's Responses to Questions

**Comments to the Author**

1. Is the manuscript technically sound, and do the data support the conclusions?

Reviewer #1: Yes

Reviewer #2: Partly

2. Has the statistical analysis been performed appropriately and rigorously? 

Reviewer #1: I Don't Know

Reviewer #2: N/A

3. Have the authors made all data underlying the findings in their manuscript fully available?

Reviewer #1: Yes

Reviewer #2: Yes

4. Is the manuscript presented in an intelligible fashion and written in standard English?

Reviewer #1: Yes

Reviewer #2: No

5. Review Comments to the Author

Reviewer #1: In this paper, the author proposes a study concerning the main characteristics of the adoption of mobile wallets. I have few comments that I think they should be addressed

- line 85: intentions to use m-wallets? -> intentions to use m-wallets?" (missing double quote at the end of research question)

- Improve quality (size, resolution, and dpi) of the figures, especially Figure 1. They are very blurry and can be very hard to understand.

- lines 435-439 you kinda explain what a neural network is, but this section should contain only the findings. You should put a reference to a good paper which explains what is a neural network, rather than including it in a section presenting your experimental results.

Reviewer #2: Authors propose a study concerning the adoption of mobile wallets. The paper is interesting, but it is not easy to follow.

First of all, what is the reason behind the choice of the three groups namely technology characteristics? Authors should motivate well this point.

Moreover, I don't think that the description of the dataset could be included in the research methodology. The title of this section leads the knowledge of the idea and how this study is structure, instead it is completely focused on the dataset. I think that this section should be revised.

The proposed model is not well defined and described, and I'm not able to understand the novelty and the differences with the state of the art. Furthermore, why the model is not compared with others in literature?

6. PLOS authors have the option to publish the peer review history of their article (what does this mean?). If published, this will include your full peer review and any attached files.

Reviewer #1: No

Reviewer #2: No

---

## [Author Response · Author response to Decision Letter 0]

15 Dec 2021

Reviewer: 1 

Comments

In this paper, the author proposes a study concerning the main characteristics of the adoption of mobile wallets. I have few comments that I think they should be addressed

- line 85: intentions to use m-wallets? -> intentions to use m-wallets?" (missing double quote at the end of research question).

- Improve quality (size, resolution, and dpi) of the figures, especially Figure 1. They are very blurry and can be very hard to understand.

- lines 435-439 you kinda explain what a neural network is, but this section should contain only the findings. You should put a reference to a good paper which explains what is a neural network, rather than including it in a section presenting your experimental results.

Authors’ Response: 

Thank you for raising this concern and bringing these points to our attention. In response, we have gone through the paper in details and have incorporated the following revisions accordingly:

1. We have added the double quotation marks at the end of research question. Please see page 5, line 106.

2. We have revised all the figures to improve the size, resolution, and dpi. The quality of updated figures is better than the earlier figures.

3. We have revised Section 4.2 and have removed explanation regarding Artificial Neutral Networks. A single sentence description has been given along with references from good research papers. Please see page 24, lines 464-466. 

Reviewer: 2

Comment

Authors propose a study concerning the adoption of mobile wallets. The paper is interesting, but it is not easy to follow.

First of all, what is the reason behind the choice of the three groups namely technology characteristics? Authors should motivate well this point.

Moreover, I don't think that the description of the dataset could be included in the research methodology. The title of this section leads the knowledge of the idea and how this study is structure, instead it is completely focused on the dataset. I think that this section should be revised.

The proposed model is not well defined and described, and I'm not able to understand the novelty and the differences with the state of the art. Furthermore, why the model is not compared with others in literature?

Authors’ Response: 

We are grateful for raising important points. Addressing these concerns will really improve the quality of our paper. In response to the above mentioned comments, we made the following revisions:

1. In response to the first comment, the motivation behind the grouping of constructs into three categories have been added to the manuscript. Please see Page 8, Lines 169-174.

2. Addressing the second comment, Table-1 (previously), containing the detailed description of the dataset, has been removed. However, a brief description about the sample has been retained in Section 3.2. Please see Pages 16, Lines 333-338.

3. To incorporate the third comments, a description about the novelty of our study is given on Pages 4-5 (Lines 84-104). To compare our model with the prior research, a literature review of 15 studies in the context of m-wallets and mobile payment has been presented in Table-1 (newly added). Please see Pages 8-10 (Table-1). Further, Section 2.2 presents detailed description of the model. 

Thank you very much for your constructive feedback. We appreciate you and all the reviewers for raising concerns regarding the various critical issues with the manuscript. In response, we have now worked diligently to address each issue and have provided greater clarity. We hope that you will approve with the additions/revisions made to address the reviewers’ concerns regarding the rigor and theoretical novelty. 

Best Regards

Authors

---

## [Decision Letter · Decision Letter 1]

10 Jan 2022

Determining the factors of M-Wallets adoption. A twofold SEM-ANN approach

PONE-D-21-29301R1

Dear Dr. Hidayat-ur-Rehman,

We’re pleased to inform you that your manuscript has been judged scientifically suitable for publication and will be formally accepted for publication once it meets all outstanding technical requirements.

Kind regards,

Barbara Guidi

Academic Editor

PLOS ONE

Additional Editor Comments (optional):

Reviewers' comments:

Reviewer's Responses to Questions

**Comments to the Author**

1. If the authors have adequately addressed your comments raised in a previous round of review and you feel that this manuscript is now acceptable for publication, you may indicate that here to bypass the “Comments to the Author” section, enter your conflict of interest statement in the “Confidential to Editor” section, and submit your "Accept" recommendation.

Reviewer #1: All comments have been addressed

Reviewer #2: All comments have been addressed

2. Is the manuscript technically sound, and do the data support the conclusions?

Reviewer #1: Yes

Reviewer #2: Yes

3. Has the statistical analysis been performed appropriately and rigorously? 

Reviewer #1: Yes

Reviewer #2: Yes

4. Have the authors made all data underlying the findings in their manuscript fully available?

Reviewer #1: Yes

Reviewer #2: Yes

5. Is the manuscript presented in an intelligible fashion and written in standard English?

Reviewer #1: Yes

Reviewer #2: Yes

6. Review Comments to the Author

Reviewer #1: The authors addressed all my comments, therefore I think the paper is now ready for publication.

Reviewer #2: The paper has been revised and all the comments have been addressed. For this reason, the paper can be accepted.

7. PLOS authors have the option to publish the peer review history of their article (what does this mean?). If published, this will include your full peer review and any attached files.

Reviewer #1: No

Reviewer #2: No

---

## [Editor Report · Acceptance letter]

20 Jan 2022

PONE-D-21-29301R1 

Determining the factors of M-Wallets adoption. A twofold SEM-ANN approach 

Dear Dr. Hidayat-ur-Rehman:

I'm pleased to inform you that your manuscript has been deemed suitable for publication in PLOS ONE. Congratulations! Your manuscript is now with our production department. 

Kind regards, 

on behalf of

Dr. Barbara Guidi 

Academic Editor

PLOS ONE